# Transcriptome Analysis of Sesame (*Sesamum indicum* L.) Reveals the LncRNA and mRNA Regulatory Network Responding to Low Nitrogen Stress

**DOI:** 10.3390/ijms25105501

**Published:** 2024-05-17

**Authors:** Pengyu Zhang, Feng Li, Yuan Tian, Dongyong Wang, Jinzhou Fu, Yasi Rong, Yin Wu, Tongmei Gao, Haiyang Zhang

**Affiliations:** 1Henan Sesame Research Center, Henan Academy of Agricultural Sciences, Zhengzhou 450002, China; zpyxiyuan@163.com (P.Z.); lifeng4090@126.com (F.L.); tianyuan322@126.com (Y.T.); wdy19860829@126.com (D.W.); fu_jinzhou@163.com (J.F.); rongyasi2022@163.com (Y.R.); wuyin628@163.com (Y.W.); 2The Shennong Laboratory, Zhengzhou 450002, China

**Keywords:** sesame, low nitrogen stress, transcriptome analysis, regulatory network

## Abstract

Nitrogen is one of the important factors restricting the development of sesame planting and industry in China. Cultivating sesame varieties tolerant to low nitrogen is an effective way to solve the problem of crop nitrogen deficiency. To date, the mechanism of low nitrogen tolerance in sesame has not been elucidated at the transcriptional level. In this study, two sesame varieties Zhengzhi HL05 (ZZ, nitrogen efficient) and Burmese prolific (MD, nitrogen inefficient) in low nitrogen were used for RNA-sequencing. A total of 3964 DEGs (differentially expressed genes) and 221 DELs (differentially expressed lncRNAs) were identified in two sesame varieties at 3d and 9d after low nitrogen stress. Among them, 1227 genes related to low nitrogen tolerance are mainly located in amino acid metabolism, starch and sucrose metabolism and secondary metabolism, and participate in the process of transporter activity and antioxidant activity. In addition, a total of 209 pairs of lncRNA-mRNA were detected, including 21 pairs of trans and 188 cis. WGCNA (weighted gene co-expression network analysis) analysis divided the obtained genes into 29 modules; phenotypic association analysis identified three low-nitrogen response modules; through lncRNA-mRNA co-expression network, a number of hub genes and cis/trans-regulatory factors were identified in response to low-nitrogen stress including *GS1-2* (glutamine synthetase 1–2), *PAL* (phenylalanine ammonia-lyase), *CHS* (chalcone synthase, CHS), *CAB21* (chlorophyll a-b binding protein 21) and transcription factors *MYB54*, *MYB88* and *NAC75* and so on. As a trans regulator, lncRNA *MSTRG.13854.1* affects the expression of some genes related to low nitrogen response by regulating the expression of MYB54, thus responding to low nitrogen stress. Our research is the first to provide a more comprehensive understanding of DEGs involved in the low nitrogen stress of sesame at the transcriptome level. These results may reveal insights into the molecular mechanisms of low nitrogen tolerance in sesame and provide diverse genetic resources involved in low nitrogen tolerance research.

## 1. Introduction

Nitrogen (N) is an essential mineral nutrient element for plant growth and development, which serves as a constituent of many important macromolecules, including proteins, enzymes and several plant hormones [1,2]. The application of chemical fertilizer, especially nitrogen fertilizer, has become one of the important ways to increase crop yield. In recent years, in order to increase crop yield, the use of nitrogen fertilizer has increased year by year, but the yield has increased slowly. The main reason is that nitrogen fertilizer cannot be completely absorbed by plants, and only 25~50% is absorbed. About 10~35% will remain in the soil, and most of the rest will be lost to the environment through nitrification and denitrification, ammonia volatilization, leaching and runoff loss, resulting in a waste of nitrogen resources. The efficiency of nitrogen fertilizer has been greatly reduced, and this agricultural production model of increasing yield by increasing nitrogen input has become unsustainable [3,4]. A better N balance or lower N input is very important for sustainable agricultural production [5]. Therefore, understanding the nitrogen utilization mechanism of sesame will help to improve the nitrogen absorption and utilization efficiency of sesame, and is of great significance for the breeding of sesame varieties with high efficiency.

Non-coding RNAs (ncRNAs) are important regulatory factors involved in plant growth, development and biotic and abiotic stress responses [6,7,8]. According to their length, ncRNAs can be classified into two groups: small ncRNAs (sRNAs, <40 nt) and long ncRNAs (lncRNAs, >200 nt) [9]. To date, a large number of lncRNAs has been identified and characterized in a variety of plant species, such as *Arabidopsis* [10], rice [11], cotton [12], soybean [13] and peach [14]. Lv et al. identified 7245 lncRNAs in maize leaves under N sufficiency or N deficiency through RNA-seq, and 637 were responsive to LN (Low Nitrogen). Co-expression analyses that suggested most of those lncRNAs are involved in energy metabolic pathways related to NADH dehydrogenase activity, oxidative phosphorylation and the N compound metabolic process [15]. Qiu et al. identified 1628 candidate lncRNAs from transcriptomic sequence under LN treatment in rice root, including 181 (89 up-regulated and 92 down-regulated) lncRNAs specifically responsive to LN treatment. Among them, LN-induced *lncRNA24320.6* was demonstrated to positively regulate the expression of *OsF3′H5*, and *lncRNA24320.6* overexpression can improve nitrogen uptake and promote rice seedling growth under LN stress [6]. Based on RNA-seq (RNA-sequence), 388 lncRNAs were detected in Populus, and 126 (118 up-regulated and 8 down-regulated) lncRNAs were significantly induced under LN stress [16]. However, no lncRNA has been identified and reported in sesame to date.

Sesame (*Sesamum indicum* L.) is one of the oldest oil crops in the world and is known as the “Queen of oils” for its high nutritional, medicinal, cosmetic and culinary qualities [17,18]. Since sesame seed protein mainly comprises N, nitrogenous fertilizers significantly impact sesame yield and seed quality [19,20]. Previous research generally focused on revealing coding genes regulated by nitrogen; in contrast, noncoding components such as lncRNAs induced by nitrogen deficiency have received little attention. In the present study, high-throughput sequencing was thus applied to analyze the expression profiles of lncRNAs and mRNAs at the sesame seedling stage under normal nitrogen (CK) and low nitrogen (LN) conditions. The integrated analysis of lncRNA and mRNA expression profile is helpful to understand the response mechanism of low nitrogen in sesame, and provides a theoretical basis for further elucidating the role of lncRNA in the process of low nitrogen stress in sesame.

## 2. Results

### 2.1. Effect of N Level on Growth Performances of Two Sesame Varieties

ZZ and MD were planted in plastic pots and treated with low nitrogen at the seedling stage. The experiment applied CK and LN stress. The changes in dry weight, N concentration and N accumulation in the CK group and LN treatment groups are shown in Table 1. Although LN treatment caused a significant reduction of shoots dry weight for the two sesame varieties, ZZ was much less affected than MD, with ZZ and MD showing 57.56% and 70.38% reduction, respectively (Table 1). Meanwhile, LN-enhanced root growth increased by 38.74% and 23.58% for ZZ and MD, respectively. The less reduction of shoot dry weight and the much increase in root dry weight both contributed to the higher relative total plant dry weight in ZZ. On the other hand, although there was little difference in shoot N concentration between the two accessions under normal N, ZZ had a significantly higher shoot N concentration than MD under LN. Under LN stress, N accumulation of shoot in ZZ was 2.16 times larger than that of MD (Table 1). Obviously, the current results proved that ZZ is more LN tolerant than MD.

### 2.2. RNA Sequencing Results

In order to investigate an appropriate time of sampling for RNA-seq analysis, we observed the expression profile of nitrogen-responsive gene *SiNRT2.1*. The transcript level of *SiNRT2.1* was gradually increased at 3d under LN stress, and then gradually decreased at 6d but still remained on a little higher level than CK. Under LN stress, the highest transcript level of *SiNRT2.1* was 6.73-fold than CK (Appendix A). The results indicated that the roots were capable of sensing the LN signal and activating relevant signal transduction at 3d after treatment, resulting in differential expression of the relevant genes, and showed highly significant differences at 9d between the N levels. So, we then took the samples at 3d and 9d h for RNA-seq analysis.

To obtain an overall view of the early LN-responsive transcriptome in the two varieties, RNA samples were prepared from the roots of both accessions at 3d and 9d after LN treatment. Gene expression profiles of the sesame roots under normal and LN conditions were analyzed. For each sample, three biological replicates were performed in sequencing. A total of 2,242,338,118 raw reads were obtained through the Illumina sequencing. After quality-control filtering, 2,233,414,890 clean reads were obtained from the tested samples (Appendix A). The Q20 and Q30 values of the clean reads were over 97.84% and 93.83%, respectively, indicating that the obtained clean reads were of high quality (Appendix A). Using the TopHat software v2.1.1, we mapped a total of 1,626,087,736 (72.92%) clean reads to the sesame genome (Appendix A).

To validate the Illumina sequencing data, nine DEGs and seven DELs were randomly selected for qRT-PCR (quantitative real-time PCR) analyses. The results from both qRT-PCR and RNA-seq analysis showed that expressions of these genes were highly consistent, thus validating the RNA-seq data (Figure 1).

### 2.3. Identification of DEGs

The transcriptional levels were normalized using the FPKM (fragments per kilobase of exon per million fragments mapped reads) method. Meanwhile, FDR (false discovery rate) of <0.05 was used as the screening threshold to test the significance of the difference in transcript abundance. In the nitrogen-sensitive line MD, 941 and 2682 DEGs were detected at 3 d and 9 d under low nitrogen stress, respectively. In the nitrogen-tolerance line ZZ, 792 and 1762 DEGs were detected at 3 d and 9 d under low nitrogen stress, respectively (Figure 2A). Consequently, we obtained a total of 3964 DEGs, which presented different LN responses between the two varieties, involving 1815 and 828 specific DEGs in MD and ZZ, respectively, whereas 1321 DEGs were common between the two varieties (Figure 2B). By comparing the mRNA expression levels at a given treatment stage between the two varieties, we identified 1396 and 3401 DEGs at 3 and 9 d of low nitrogen treatment (Figure 2B).

### 2.4. GO (Gene Ontology) and KEGG (Kyoto Encyclopedia of Genes and Genomes) Enrichment Analysis of LN Tolerance-Related DEGs

Of the total 3964 DEGs, 1227 DEGs, showing significant up-regulation in ZZ, but down-regulation or unchanged in MD, or little change in ZZ but down-regulation in MD, were selected for further investigation. Based on hierarchical clustering analysis, these DEGs could be mainly grouped into four classes (Appendix A). GO functional enrichment analysis was performed to classify these DEGs into their corresponding biological process (BP), molecular function (MF) and cellular component (CC) (Figure 3, Appendix A). In the molecular function category, DEGs with oxidoreductase activity (GO:0016491), cellulose synthase activity (GO:0016759), glucosidase activity (GO:0015926), tetrapyrrole binding (GO:0046906) and antioxidant activity (GO:0016209) were significantly enriched in the low nitrogen stress plants. In the cellular component category, DEGs related to the extracellular region (GO:0005576), cell wall (GO:0005618), cell periphery (GO:0071944) and external encapsulating structure (GO:0030312) were significantly enriched. In the biological process category, DEGs involved in response to carbohydrate catabolic process (GO:0016052), immune effector process (GO:0002252), phenylpropanoid biosynthetic process (GO:0009699), hormone-mediated signaling pathway (GO:0009755) and phloem or xylem histogenesis (GO:0010087) were significantly enriched. In addition, DEGs in response to the phenylpropanoid metabolic process (GO:0009698) were also affected.

In addition to GO analysis, 1227 DEGs were mapped to terms in KEGG pathway enrichment, and the encoded enzymes were assigned to 103 KEGG pathways (Appendix A and Appendix A), including carbohydrate, amino acid, terpenoids and polyketides, lipid, energy and other metabolisms. Among these pathways, DEGs involved in phenylpropanoid biosynthesis (31), plant hormone signal transduction (23), glutathione metabolism (14), and flavonoid biosynthesis (7) were significantly enriched.

### 2.5. Identification of DELs

By removing potential protein-coding transcripts (PCTs), a total of 8399 common transcripts were defined as lncRNAs, including 1800 known lncRNAs and 6599 novel lncRNAs (Figure 4A). Among novel lncRNAs, 743 (11.26%), 2245 (34.02%), 619 (9.38%) and 185 (2.80%) belonged to sense-lncRNAs, antisense lncRNAs, intronic lncRNAs and intergenic-lncRNAs, respectively (Figure 4B). In comparison with PCTs, 6252 (94.74%) of the novel lncRNAs had fewer (≤3) exons and 5951 (90.18%) had shorter (≤2000 bp) tags (Figure 4C,D).

The DELs were identified according to the criteria |log_2_ FC| > 1, FDR < 0.05. In the nitrogen-sensitive line MD, 40 and 146 DELs were detected at 3d and 9d under low nitrogen stress, respectively. In the nitrogen-tolerance line ZZ, 37 and 57 DELs were detected at 3d and 9d under low nitrogen stress, respectively (Figure 5A). A total of 224 DELs were identified between different treatment groups. Among them, 5 DELs were tested in four treatment groups, 144 and 53 DELs were specifically responsive to low nitrogen stress in MD and ZZ, respectively. By comparing the lncRNA expression levels at a given treatment stage between the two varieties, we identified 71 and 180 DELs at 3 and 9 d of low nitrogen treatment (Figure 5B).

### 2.6. Identification and Enrichment Analyses of Target Genes of DELs

In order to further clarify the roles of the 224 potential low nitrogen stress-responsive DELs, we identified from all the DEGs the potential cis- and trans-target transcripts. In total, 209 lncRNA-mRNA pairs including 21 trans- and 188 cis-pairs were detected, which were speculated to be involved in response to low nitrogen stress in sesame root. Among them, 172 DEGs were located between the 100 kb upstream and downstream of the 224 DELs and were significantly correlated (Pearson correlation coefficient, |PCC| > 0.6, *p* < 0.05) with their corresponding DELs, which were thereby defined as the cis-regulated target transcripts (Appendix A). These 172 cis-transcripts were regulated by 92 DELs. Meanwhile, 21 DEGs with free energy < −0.2, |PCC| > 0.8 and *p* < 0.01 were identified as the trans-target transcripts of 17 DELs (Appendix A).

KEGG enrichment analysis indicated that “Glycosaminoglycan degradation (ko00531)”, “Cysteine and methionine metabolism (ko00270)” and “Phenylpropanoid biosynthesis (ko00940)” pathways were significantly enriched with the target transcripts (Appendix A). GO analysis uncovered 10 (membrane part, intrinsic component of membrane, plastid part and others), 19 (UDP-glucosyltransferase activity, glucosyltransferase activity, tetrapyrrole binding and others), and 8 (alkaloid metabolic process, glycine metabolic process, response to stress and others) terms as the most significantly enriched GO terms in cellular component (CC), molecular function (MF), and biological process (BP), respectively (Appendix A).

### 2.7. WGCNA Identifies Candidate Modules Associated with Low Nitrogen Response

The WGCNA was used to analyze the connection between genes and physiological traits, discovering the hub genes associated with physiological and biological traits. The soft-threshold power of β was determined as 6 when the scale-free topology index was 0.9 (Appendix A). Finally, 23,162 genes were assigned to 29 distinct modules built with the parameters (deepSplit = 2 and minModuleSize = 50), which were labeled with different colors (Figure 6A). The number of genes in each module ranged from 64 to 7488, and 20,609 (88.98%) genes were classified into the top ten modules (Appendix A). As shown in Figure 6B, we successfully identified three modules (brown4, brown and darkgreen) significantly (r > |0.52|) associated with physiological or biological traits of two varieties of seedlings under low nitrogen stress, which were identified as key modules.

In order to determine the biological function of DEL target genes in each co-expression module, we carried out a KEGG pathway enrichment analysis (Appendix A). The transcripts in the brown, brown4 and darkgreen modules were significantly enriched in four (Photosynthesis, Carbon fixation in photosynthetic organisms, Carbon metabolism and Biosynthesis of amino acid), four (Phenylalanine, tyrosine and tryptophan biosynthesis, Cysteine and methionine metabolism, Plant hormone signal transduction and Glycerolipid metabolism) and five (Stilbenoid, diarylheptanoid and gingerol biosynthesis, Flavonoid biosynthesis, Oxidative phosphorylation, Plant-pathogen interaction and Phenylpropanoid biosynthesis) pathways, respectively (Appendix A).

### 2.8. Regulatory Network Mediated by LncRNAs and Their Target Genes

Gene co-expression networks can be used to analyze gene functions with biological processes, prioritize candidate genes or discern transcriptional regulatory programs [21] The weight value indicates the correlation of gene relationship pairs in the module, one relationship pair for every two genes. We selected the top 200 relationship pairs in terms of weight to draw the network regulatory map, and then selected the five genes with the highest degree of connectivity as the hub genes from them. The correlation network of the brown4 module is shown in Figure 7, which contains 15 DELs, 12 TFs, and 47 DEGs. Notably, the MYB transcription factor MYB54 (LOC105157332) was identified as a hub (KME = 0.94, TOM = 0.37) gene in the module, which was trans-regulated by the lncRNA MSTRG.13854.1. The top five co-expressed mRNAs in brown4 module included LOC105165713 (glutamine synthetase, GS1-2), LOC105161317 (phenylalanine ammonia-lyase, PAL), LOC105171854 (chalcone synthase, CHS), LOC105173574 (gamma-glutamyl transpeptidase 3, GGT3) and LOC105162027 (NAC transcription factor, NAC75). In the brown module, the co-expression network diagram includes three DELs, three TFs and thirty-six DEGs. The top five co-expressed mRNAs in brown4 module included LOC105178077 (photosystem I reaction center subunit N, PSAN), LOC105167309 (chlorophyll a-b binding protein 21, CAB21), LOC105177713 (oxygen-evolving enhancer protein, PSBO), LOC105158872 (glyceraldehyde-3-phosphate dehydrogenase, GAPB) and LOC105156552 (MYB transcription factor, MYB88) (Appendix A).

## 3. Discussion

Nitrogen is considered to be the single most important factor for determining crop productivity and grain quality, because it is the most mineral nutrient required by plants [22]. Unfortunately, nitrogen uptake efficiency is very low, and only 40% of the applied N is absorbed by fertilized crops [23]. Therefore, it is necessary to develop crop cultivars with high LN tolerance or NUE to cope with the issue. Previous studies showed that there are significant differences in low nitrogen tolerance among species or genotypes within a species. Dry weight or relative biomass is often used as an indicator of plant tolerance to low nutrition stress [24,25]. In this study, the differences in growth performance between the two sesame varieties confirmed our result as previously described that ZZ is more tolerant than MD under low nitrogen stress, and ZZ had a higher N absorption and translocation capability than MD (Table 1).

LncRNAs are known to play vital roles in stress response, such as low nitrogen, salinity or pathogenic infection [6,7,26]. Through whole genome RNA-seq, 4373 lncRNAs in *Arabidopsis thaliana*, 2788 lncRNAs in *rice*, 12,817 lncRNAs in *Zea mays* and approximately 12,000 lncRNAs in *Brassica napus* were identified [7,27,28,29]. To date, only a few lncRNAs have been functionally characterized. For example, in continuous red light, the *Arabidopsis thaliana* lncRNA HID1 (HIDDEN TREASURE 1) promotes photomorphogenesis by inhibiting the promoter activity of PIF3 (PHYTOCHROME-INTERACTING FACTOR 3) [30]. Overexpression of T5120 in *Arabidopsis* promoted the response to nitrate, enhanced nitrate assimilation and improved biomass and root development [31]. However, the role of LncRNAs in regulating nitrogen use efficiency in sesame remains to be studied.

To investigate how low nitrogen treatment affects lncRNA expression in sesame, we analyzed the expression of DELs and DEGs in low nitrogen treatment and normal sesame roots through RNA-seq. In this study, a total of 3964 DEGs have been identified (Figure 2). Among them, 1227 low nitrogen tolerance DEGs were selected for further investigation. Moreover, through the prediction of four software, a total of 6599 high-confidence lncRNAs were identified (Figure 4). Compared with protein-coding genes, lncRNAs were shorter in length and had fewer exons in structures, which were similar to previous findings [32,33]. Subsequently, we executed the comparative transcriptomic analysis, 224 differential DELs were identified between two varieties under low nitrogen stress. Most nitrogen-responsive DELs exhibit different expression patterns in the early stages of low nitrogen stress, especially in nitrogen-sensitive line MD (Figure 5). Meanwhile, we identified 188 cis- and 21 trans-DELs-DEGs pairs (Appendix A). Among them, DELs MSTRG.17494.1 targets three plant NRT1/PTR family members NPF2.11(LOC105168760), NPF2.10 (LOC105168762) and NPF4.6 (LOC105168767), a previously study has proved that the NRT/NPF protein play vital roles in moving substrates including amino acids, peptides, nitrate, dicarboxylates, glucosinolates, ABA, IAA, JA and so on [34]. To further recognize the function of these DELs under low nitrogen stress, we performed the KEGG pathway and GO term enrichment analysis for the target transcripts of the DELs (Appendix A). Some low nitrogen stress-responsive GO terms and pathways such as “glucosyltransferase activity” term and “Phenylpropanoid biosynthesis” pathway [35,36] were significantly enriched with the target transcripts. These findings suggested that the DELs were involved in the low nitrogen response of sesame seedlings and contributed to the difference in nitrogen tolerance between these two varieties.

Recent progress in transcriptomics and co-expression networks have enabled us to predict the inference of the biological functions of genes with the associated environmental stress. Gene co-expression networks are a systems biology method for capturing transcriptional patterns and predicting gene interactions into functional and regulatory relationships [37]. Based on the analysis of RNA-seq and gene co-expression network under drought stress, nine key genes of drought response were found, which could lead to a better understanding of drought tolerance and facilitate the breeding of drought-resistant peanut cultivars [38]. Furthermore, Wang et al., through weighted gene co-expression analysis, pinpointed important key pathways and hub genes under saline–alkaline stress, which can significantly enhance useful knowledge for improving yield and quality under saline–alkaline conditions in sorghum [39]. Phenylalanine is a key amino acid at the interphase of primary and secondary metabolism, and PAL is an initial rate-limiting enzyme in phenylpropanoid synthesis. PAL acts as a key enzyme in response to various stress, such as low temperature, salt, low nitrogen, phosphate and iron [40]. Therefore, PAL has generally been considered one of the main markers of environmental stress [41]. The photosynthetic capacity of plants decreased significantly under LN stress, and was vulnerable to oxidative damage caused by excessive light [42]. As an adaptive strategy, the synthesis of photoprotective pigments such as flavonols in plants exposed to LN can filter harmful radiation wavelengths and avoid damage to plants [43]. In this study, we identified eleven core genes in response to low nitrogen stress through weighted gene co-expression analysis. Among them, DEGs (LOC105161317 and LOC105171854) encoding the two key enzymes phenylalanine ammonia-lyase (PAL) and chalcone synthase (CHS), respectively, are involved in the phenylalanine metabolism and flavonoid biosynthesis pathway. It may be assumed that the enhanced phenylpropanoid metabolism and flavonoid biosynthesis under LN stress may contribute to its high tolerance.

Several TFs have been reported to be involved in the regulation of nitrogen metabolism. Previous studies showed that R2R3-type MYB TF, *CmMYB1*, enhances the expression of *CmNRT*, *CmNAR*, *CmNIR*, *CmAMT* and *CmGS* under N stress [29]. *SiMYB3* can regulate root development by regulating plant root auxin synthesis under low-nitrogen conditions [44]. In wheat, the NAC transcription factor *TaNAC2-5A* significantly promotes root growth and enhances the expression of NRT1 and NRT2 families, and thus, increasing N uptake and grain yield in wheat [45]. In our study, *MYB54* (LOC105157332), *MYB88* (LOC105156552) and *NAC75* (LOC105162027) were identified as key transcription factors in response to low nitrogen stress. And MYB54 was trans-regulated by the lncRNA *MSTRG.13854.1*. These results indicate the potential regulatory functions of the predicted transcription factor in sesame’s response to low nitrogen stress.

The identification and analysis of differential genes in response to low nitrogen stress in sesame seedlings are helpful in revealing the genetic mechanism of low nitrogen tolerance. In this study, the RNA-seq technique was used to compare the difference in transcriptional level in response to low nitrogen stress between Zheng Zhi HL05 and Burmese high-yielding roots. Based on the above research results, we proposed the possible molecular mechanism of low nitrogen tolerance of sesame Zhengzhi HL05. Firstly, increase nitrate absorption directly by up-regulating the expression of genes related to nitrate absorption and transport. Secondly, obtain more nitrogen sources by regulating the expression of growth and development-related genes, such as plant hormones and kinases. Finally, it also adapts to low nitrogen stress by regulating key transcription factors related to nitrogen signals to increase nitrogen uptake in roots (Figure 8). In addition, this study combined with WGCNA also identified some key genes related to low nitrogen tolerance. Subsequently, we will select key genes, such as NPF4.6 (LOC105168767), MYB54 (LOC105157332), PAL (LOC105161317) and so on to verify gene function and participate in the molecular mechanism of nitrogen response, which can provide genetic resources for low nitrogen tolerance breeding.

## 4. Materials and Methods

### 4.1. Plant Materials and Treatment

The experiment was carried out in a greenhouse at Henan Academy of Agricultural Sciences, China. Sesame seeds of two genotypes Zhengzhi HL05 (ZZ, nitrogen efficient) and Burmese prolific (MD, nitrogen inefficient) [46] were surface-sterilized with 3.0% NaCLO for 15 min, rinsed thoroughly with distilled water 5 times, and then soaked for 24 h in distilled water at 30 °C in the dark. Then the seeds were sown in soil and vermiculite mixture (3:1) in a greenhouse at 30 °C/22 °C with 14 h/10 h light/dark photoperiod cycles, and a light intensity of 350 µmol·m^−2^·s^−1^, 70% relative humidity. When seedings had two fully expanded leaves, the similar seedlings were transferred into a nutrient solution for nitrogen treatment in black plastic pots (4L). The pH of the solution was adjusted to 7.0 with NaOH, and replaced every 4 days. The nitrogen treatment with 0.2 mM N was LN treatment (T), and that with 17.6 mM N was the control (CK).

For biomass and N content determination, the seedlings were harvested and separated into shoots and roots, at 14 d after LN treatment. All the plant samples were heated at 105 °C for 30 min, dried at 80 °C until their weight remained constant, and then dry weight was recorded. N content was determined using Foss Kjeltec 8400.

In order to know the time course of gene *SiNRT2.1* expression under LN stress, the roots of ZZHL05 were harvested with three biological replicates at 0 d, 3 d, 6 d, 9 d and 12 d after LN treatment, frozen in liquid nitrogen immediately, and stored at −80 °C for use in RNA extraction.

The roots of ZZ and MD were harvested at 3 d and 9 d after LN treatment, denoted as ZZ-T3d, ZZ-T9d, MD-T3d and MD-T9d, and the control groups were named ZZ-CK3d, ZZ-CK9d, MD-CK3d and MD-CK9d, respectively. Three biological replicates were set for each group (each biological replicate contained five individuals). All fresh samples were stored at −80 °C until physiological assays and transcriptome sequencing.

### 4.2. RNA Extraction and Library Construction for Illumina Sequence

The total RNA of each sample was isolated using a TRIzol total RNA extraction kit (Invitrogen, Carlsbad, CA, USA) according to the manufacturer’s instructions. RNA quality was characterized on an agarose gel electrophoresis and NanPhotometer^®^ spectrophotometry (IMPLEN, Westlake Village, CA, USA). High-quality RNA with 28S:18S more than 1.5 and absorbance 260/280 ratios between 1.8 and 2.0 was used for library construction and sequencing.

Sequencing library construction was performed using the Illumina TruSeq™ RNA Sample Preparation Kit (TaKaRa, Dalian, China) according to the manufacturer’s instructions. Initially, magnetic beads with poly-T oligos were used to purify mRNA from the total RNA, and the purified mRNA was cleaved into short fragments using NEBNext First Strand Synthesis Reaction Buffer (5×) under high temperature. Then the fragments were used as templates to synthesize first-stand cDNA using SuperScript II (TaKaRa, Dalian, China). Illumina paired-end sequencing adapters were ligated for preparation of hybridization after making adenylation of the 3′ ends of DNA fragments. AMPure XP system was used to purify the library fragments and preferentially select cDNA fragments (Beckman Coulter, Beverly, MA, USA). With ligated adapters on both ends, DNA fragments were selectively amplified and enriched. Then, PCR products were purified again using an AMPure XP system and quantified using the Agilent Bioanalyzer 2100 system. Finally, libraries were sequenced on an Illumina HiSeq 4000 platform (Guangzhou Genedenovo Biotechnology Co., Guangzhou, China).

### 4.3. Data Filtering and Mapping Reads to the Genome

The raw reads were generated through the Illumina data processing pipeline (version 1.8). For further analysis, Bowtie 2 was used to filter out rRNA [47]. Then, the clean reads were obtained by removing reads containing adapter or poly-N. Meanwhile, the Q20, Q30, GC contents, and sequence duplication level of the clean data were calculated. All the subsequent analyses were conducted using high-quality clean reads. Subsequently, the sesame genome sequence and annotation data were downloaded, and the resulting paired-end clean reads were aligned to the reference genome on TopHat v2.1.1 software (http://tophat.cbcb.umd.edu/ accessed on 30 April 2022) [48]. In addition, Trinity (v2.15.1) software was used to de novo assemble the mapped transcripts [49]. The annotated transcripts were then used for further analyses.

### 4.4. Identification of LncRNAs

The assembled transcriptome data were further screened to identify lncRNAs. First, all assembled transcripts were combined using Cuffmerge software in the Cufflinks Support Package (v2.2.1), and those with read coverage of less than three and lengths less than 200bp were removed. Subsequently, transcripts with annotated sesame mRNAs were compared using Cuffcompare software in the Cufflinks Support Package (v2.2.1), and then the transcripts that were overlapped with the sesame mRNAs were removed. Finally, transcripts that showed potential coding capabilities through assessments with four software, namely, CNCI (Coding-Non-Coding Index, 0.9-r2) [50], CPC (Coding Potential Calculator, version 2) [51], FEElnc [52] and Pfam 28.0 [53], were removed. Finally, the remaining transcripts were identified as lncRNAs.

### 4.5. Differential Expression Analysis of Genes LncRNAs

The expression levels of unigene were normalized, and the FPKM was calculated by the Cuffdiff (v2.2.1) software [54]. Differentially expressed genes (DEGs) and lncRNAs (DELs) between control and low nitrogen treatment (three biological replicates per time point) were analyzed using the DESeq R package (1.10.1) [55]. The |log_2_ (Fold Change, FC)| > 1 and FDR < 0.05 were the thresholds for screening DEGs and DELs [56].

### 4.6. GO Term and KEGG Pathway Enrichment Analysis

Kyoto Encyclopedia of Genes and Genomes (KEGG) pathway and gene ontology (GO) enrichment analyses were performed using OmicShare, a free online platform for data analysis (www.omicshare.com/tools accessed on 21 August 2023).

### 4.7. Prediction of LncRNA Targets

To identify the cis-target transcripts of lncRNAs, we searched for DEGs between 100 kb upstream and downstream of the lncRNAs [57]. The Pearson correlation coefficient (PCC) between the lncRNA and the corresponding DEGs was then calculated based on their expression levels. The DEGs that met the strict standards (|PCC| > 0.6, *p* < 0.05) were considered as cis-target transcripts of the lncRNAs. Furthermore, the LncTar program was used to predict the trans-targets of lncRNAs based on complementary base pairing [58]. The transcript was considered a trans target of the lncRNA when the free energy of pairing sites between transcript and lncRNA was lower than the threshold of standardized free energy (ndG < −0.2) [33]. In addition, the PCC between the lncRNA and the corresponding transcript was calculated. Those mRNAs with |PCC| > 0.8 and *p*-value < 0.01 in lncRNA-mRNA pairs were defined as the putative trans-target mRNAs of the lncRNAs [59].

### 4.8. Weight Gene Co-Expression Network Construct and Analysis

The WGCNA was executed with the WGCNA (v1.69) package in R [60] based on the normalized expressions of all genes. Then, the Pearson correlation analysis was used to identify modules significantly (Pearson’s correlation coefficient > |0.53|) associated with physiological or biological traits of sesame seedlings under low nitrogen stress. The DEL and DEG co-expression networks were constructed according to the co-expression associations between DELs and DEGs, and were visualized using the Cytoscape 3.2.7 software (http://cytoscape.org/, accessed on 9 November 2023).

### 4.9. Validation of RNA-Seq Analysis through qRT-PCR

qRT-PCR analysis was performed to verify the sequencing results, 9 DEGs and 7 DELs were randomly selected for qRT-PCR. SiUBQ6 gene was used as the internal reference gene [61]. The Model CFX96 Connect Real-Time System (Bio-Rad, Hercules, CA, USA) was used for real-time fluorescence quantitative PCR (qRT-PCR). The 20 μL reaction volume contained 1 μL of diluted cDNA, 1 μL of forward and reverse primers (10 μM), 10 μL of 2× SYBR^®^ Premix ExTaq II (TaKaRa, Dalian, China) and 7 μL of ddH_2_O. The PCR amplification was performed at 95 °C for 5 min, followed by 40 cycles of 95 °C for 15 s, 60 °C for 15 s, then followed by 72 °C for 20 s. The relative quantification was calculated by the 2^−ΔΔCT^ method [62]. Six independent biological replicates were designed here. The primer sequences used in the qRT-PCR assay are listed in Appendix A.

### 4.10. Statistical Analysis

Significant differences in physiological traits and gene expression among treatments and genotypes were tested using the Duncan’s Multiple Range Test (DMRT) on data processing system (DPS, v20.00) statistical software, and the differences at *p* < 0.05 and *p* < 0.01 were considered significant and highly significant, respectively. All experimental data were expressed as mean ± standard deviation (SD).

## 5. Conclusions

In conclusion, we identified 6355 lncRNAs from two sesame varieties with contrasting nitrogen efficiency by bioinformatic analysis. In total, 3964 DEGs and 221 DELs were identified in response to low nitrogen stress. Combining WGCNA and co-expression analysis, a total of 11 consistently low nitrogen-responsive candidate genes and lncRNAs were identified. Further research on biological function, including GO enrichment and KEGG analysis, should provide useful information for obtaining a deeper understanding of the mechanisms of lncRNA regulation during the early stages of nitrogen deficiency in sesame seedling development, and for providing new insights enabling increased efficiency of breeding for nitrogen utilization.

## Figures and Tables

**Figure 1 ijms-25-05501-f001:**
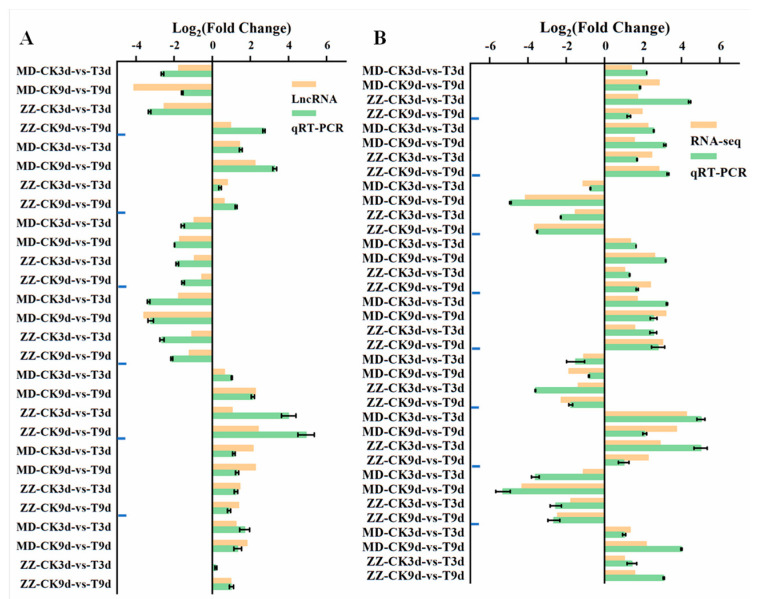
Quantitative real-time PCR validation of LN responsive lncRNAs and genes. The bars represent SE (*n* = 6). (**A**) Relative expression levels of lnRNAs. From top to bottom is MSTRG.4330.1, MSTRG.5300.1, MSTRG.5563.1, MSTRG.7134.2, MSTRG.10253.1, MSTRG.20071.1 and MSTRG.24385.1. (**B**) Transcript levels of 9 DEGs and the corresponding expression data of RNA-Seq. From top to bottom is LOC105168587, LOC105159638, LOC105169455, LOC105169857, LOC105155458, LOC105157670, LOC105156385, LOC105167240 and LOC105170941.

**Figure 2 ijms-25-05501-f002:**
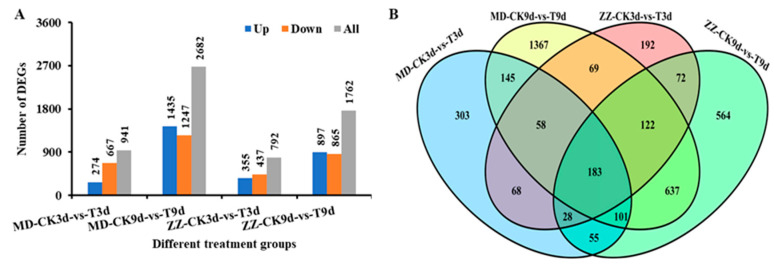
Statistics of DEGs in different treatment groups under low nitrogen stress. (**A**) The number of DEGs from different treatment groups. (**B**) Venn diagram shows the numbers of DEGs in different comparison groups.

**Figure 3 ijms-25-05501-f003:**
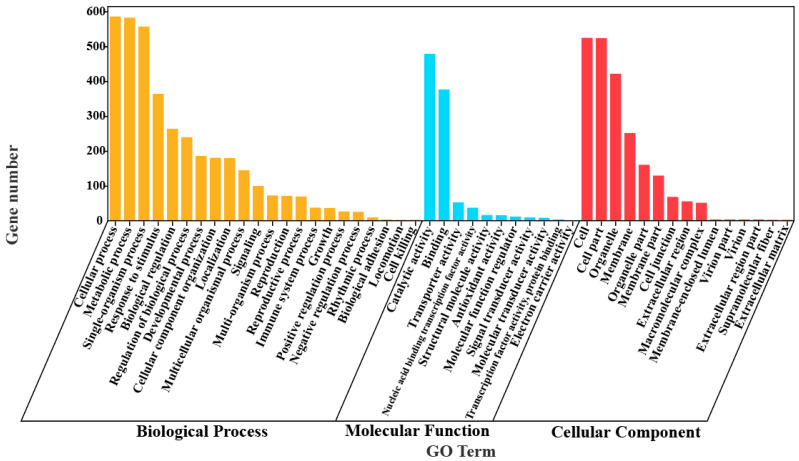
GO function enrichment analysis of LN tolerance-related DEGs.

**Figure 4 ijms-25-05501-f004:**
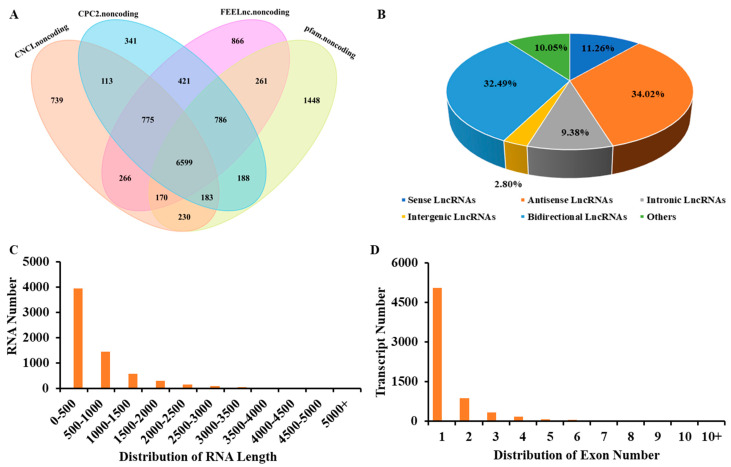
Prediction of lncRNAs with different programs. (**A**) The Venn diagram in the figure represents the prediction of lncRNAs using different software. (**B**) Bar plot shows the counts of different types of predicted lncRNAs. (**C**) The transcript length of novel lncRNAs. (**D**) The exon number of novel lncRNAs.

**Figure 5 ijms-25-05501-f005:**
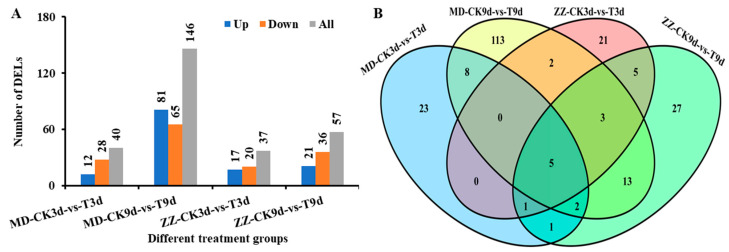
Statistics of DELs in different treatment groups under low nitrogen stress. (**A**) The number of DELs from different treatment groups. (**B**) Venn diagram shows the numbers of DELs in different comparison groups.

**Figure 6 ijms-25-05501-f006:**
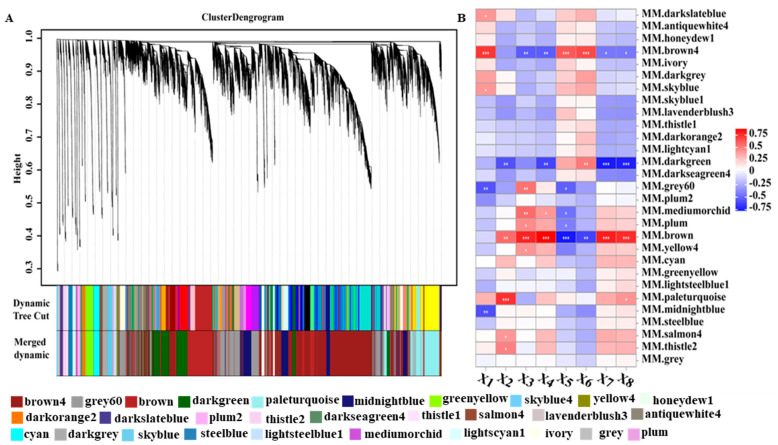
The co-expression analysis of low nitrogen stress-responsive genes. (**A**) The gene expression network of low nitrogen response genes was analyzed through WGCNA, and the genes were clustered into different co-expression modules. (**B**) Correlation analysis was carried out between the co-expression modules of various genes under low nitrogen conditions and physiological indexes related to low nitrogen stress. X1: taproot length (cm); X2: root dry weight (mg/plant); X3: root nitrogen content (%); X4: root nitrogen accumulation (mg/plant); X5: root nitrogen utilization efficiency (mg/mg); X6: root nitrogen absorption efficiency (%); X7: nitrate reductase activity (μg/(g·h)); X8: glutamine synthetase activity (μmol/(g·h)). *: *p* < 0.05, **: *p* < 0.01, ***: *p* < 0.001.

**Figure 7 ijms-25-05501-f007:**
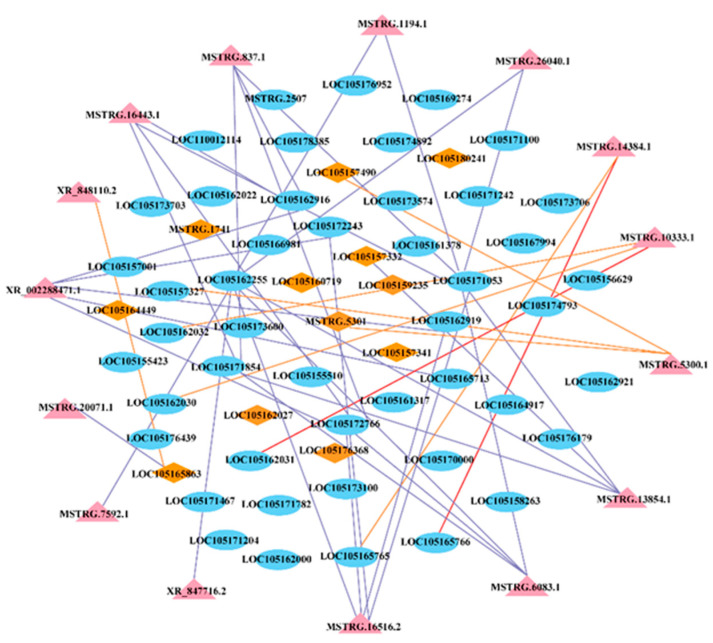
Regulatory network mediated by lncRNAs and their target genes in brown4 module. The light blue oval represents the gene, the pink triangle represents lncRNA, and the orange diamond represents the transcription factor. LncRNA-mRNA regulation: yellow lines indicate cis-regulated, red lines indicate antisense-regulated, and purple lines indicate trans-regulated.

**Figure 8 ijms-25-05501-f008:**
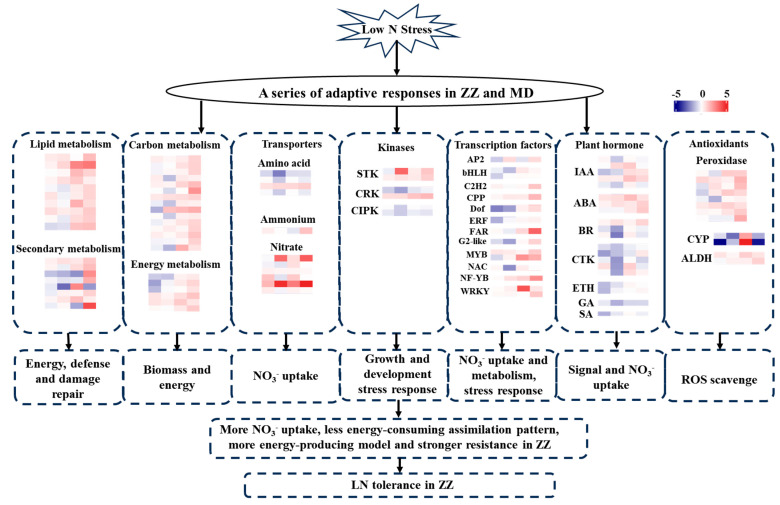
A hypothetical model of low N tolerance mechanism underlying in Zhengzhi HL05. Genes are shown by different colors and log_2_ (FC) of T- vs. CK are shown in a color gradient from low (blue) to high (red). For each heatmap from left to right: MD-3 d (first column), MD-9 d (second column), ZZ-3 d (third column), ZZ-9 d (fourth column).

**Table 1 ijms-25-05501-t001:** Growth performances of two sesame varieties at 14 d after low N treatment.

Traits	Tissue	ZZ			MD		
		CK	LN	LN/CK	CK	LN	LN/CK
Dry weight (mg/plant DW)	Shoot	448.50 ± 13.37 a	190.33 ± 8.72 c	0.42	346.67 ± 10.24 b	102.67 ± 3.26 d	0.30
	Root	65.95 ± 4.10 bc	91.50 ± 2.87 a	1.39	58.10 ± 7.26 c	71.80 ± 4.75 b	1.24
N concentration (%)	Shoot	7.07 ± 0.03 a	3.94 ± 0.04 c	0.56	6.51 ± 0.02 b	3.37 ± 0.20 d	0.52
	Root	4.61 ± 0.15 a	3.14 ± 0.19 c	0.68	3.99 ± 0.17 b	2.32 ± 0.25 f	0.58
N accumulation (mg/plant DW)	Shoot	31.69 ± 1.05 a	7.49 ± 0.33 c	0.24	22.56 ± 0.73 b	3.46 ± 0.32 d	0.15
	Root	3.04 ± 1.11 a	2.87 ± 0.09 a	0.95	2.32 ± 0.30 b	1.67 ± 0.13 c	0.72

CK: normal N level (17.86 mM N); LN: low N level (0.2 mM N); LN/CK: relative. For each line, different lowercase letters indicate significant differences (*p* ≤ 0.05) among the treatments and genotypes, *n* = 6.

## Data Availability

Raw reads of Illumina RNA-seq generated in this study are available from the Sequence Read Archive (SRA) at the National Center for Biotechnology Information (NCBI) under Project ID PRJNA936595.

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
