# Peer review of "Transcriptome Analysis of Sesame (Sesamum indicum L.) Reveals the LncRNA and mRNA Regulatory Network Responding to Low Nitrogen Stress"

_ijms, 2024, doi:10.3390/ijms25105501_

Round 1
Reviewer 1 Report
Comments and Suggestions for Authors
The manuscript with the title “Transcriptome analysis of sesame (Sesamum indicum L.) reveal the LncRNAs and mRNAs regulatory network responding to low nitrogen stress”, investigated how low nitrogen treatment effects lncRNA expression in sesame by analyzing the expression of DELs and DEGs in low nitrogen treatment and normal sesame roots by RNA-seq. Authors found out that DELs were involved in low nitrogen response of sesame seedlings and contributed to the difference of nitrogen tolerance between these two varieties.
Note: in some places authors used separation of thousands “,” while in other places they did not. Please be consistent – either use it or not in the whole manuscript (e.g. Lines 306-307).
Line 265 – moule, please check spelling
Line295 - Arabidopsis thaliana, italic
The manuscript has a good degree of novelty and represents a good preliminary study on mechanisms of low N tolerance in sesame during early growth.
General remark
The study is relevant for early growth stages. The study was also undertaken with the purpose to help breeding programs. Indeed, seedling establishment is critical and the study was conducted on plants in this growth stage. However, plant transcriptome is different across growth stages. Particularly, mechanisms at play during seed filling would be very insightful.
I suggest authors to broaden the discussion on possible future steps proposed for increasing our insight on successful biotechnologies for the cultivation of sesame under low N. What the next possible research steps shall be?
Best regards.
Comments on the Quality of English Languageminor English style and grammar revision is advised.
A few spelling mistakes.
Author Response
Dear Editor,
Thank you so much for your hard work on our manuscript. According to the reviewer’s comments, we carefully revised the manuscript “Transcriptome analysis of sesame (Sesamum indicum L.) reveal the LncRNAs and mRNAs regulatory network responding to low nitrogen stress” as follows:
Reviewer comments:
Reviewer #1:
The manuscript with the title “Transcriptome analysis of sesame (Sesamum indicum L.) reveal the LncRNAs and mRNAs regulatory network responding to low nitrogen stress”, investigated how low nitrogen treatment effects lncRNA expression in sesame by analyzing the expression of DELs and DEGs in low nitrogen treatment and normal sesame roots by RNA-seq. Authors found out that DELs were involved in low nitrogen response of sesame seedlings and contributed to the difference of nitrogen tolerance between these two varieties.
Note: in some places authors used separation of thousands “,” while in other places they did not. Please be consistent – either use it or not in the whole manuscript (e.g. Lines 306-307).
Response: Thanks for the reviewer’s advice, we used separation of thousands in line17-18,57,61,137,138,140,141,151,169,179,180,181,183,184,231,234,298,299,310,311,509,510.
Line 265 – moule, please check spelling
Response: Thanks for the reviewer’s advice, we made the revision in line 266.
Line295 - Arabidopsis thaliana, italic
Response: Thanks for the reviewer’s advice, we made the revision in line 299-302.
The manuscript has a good degree of novelty and represents a good preliminary study on mechanisms of low N tolerance in sesame during early growth.
General remark
The study is relevant for early growth stages. The study was also undertaken with the purpose to help breeding programs. Indeed, seedling establishment is critical and the study was conducted on plants in this growth stage. However, plant transcriptome is different across growth stages. Particularly, mechanisms at play during seed filling would be very insightful.
Response: We are grateful for your good suggestion. In the follow-up study, we will carry out field experiments to analyze the physiological response and molecular mechanism of high nitrogen efficiency in sesame by measuring dry matter accumulation, tissue nitrogen content, photosynthetic rate, chlorophyll content in leaves, activities of enzymes related to nitrogen metabolism, yield and its components, combined with transcriptome, proteomics and metabonomics analysis.
I suggest authors to broaden the discussion on possible future steps proposed for increasing our insight on successful biotechnologies for the cultivation of sesame under low N. What the next possible research steps shall be?
Response: We are grateful for your good suggestion. In this study, through RNA-seq and WGCNA analysis, we identified a number of key candidates for low nitrogen response lncRNA, core genes and transcription factors. In the follow-up study, we will construct overexpression and knockout vectors, transform sesame seeds to obtain transgenic plants, and verify the function of lncRNA or key genes under low nitrogen stress by measuring physiological and biochemical indexes. The molecular mechanism of low nitrogen tolerance of key genes was analyzed by DAP-seq (TFs), Y1H or Y2H and other tests, which can provide valuable insights into the roles of lncRNAs and mRNA in the sesame response to nitrogen starvation.
Reviewer 2 Report
Comments and Suggestions for Authors
I congratulate the authors for their work and manuscript. Manuscript ID: ijms-2977186 "Transcriptome analysis of sesame (Sesamum indicum L.) reveal the LncRNAs and mRNAs regulatory network responding to low nitrogen stress by Zhang and collaborators address the important issue of nitrogen assimilation and stress. Using sesame as a crop of choice, the work involves transcriptome analysis of protein coding and non-coding genes. Network mapping assisted the identification of important genes involved in nitrogen metabolism. Interestingly, interacting pairs of protein coding and non-coding RNAs were also identified, showing trans-regulation. The manuscript is well written and methods detailed. Results are presented clearly and support discussion and conclusion. It should be interesting to IJMS readers providing novel information on nitrogen metabolism and low nitrogen tolerance. I have only minor suggestions listed below and will recommend acceptance after minor edits to give you the opportunity to work on them.
In the title, use "reveals" instead of "reveal".
line 24: suggestion "...factors were identified in response to low-nitrogen stress...".
line 50-51: review sentence for clarity.
line 303-305: review sentence for clarity.
line 316: suggestion "...NPF4.6 (LOC105168767). A previous study...".
Table 1: Authors mention on legend "For each line, different low-96 ercase letters indicate significant differences (P≤0.05) among the treatments and genotypes", but no letters are shown next to the values on Table 1, please check.
Figure 1: Add number of replicate measurements on legend, and include error bars for fold change measurements.
Lines 237-239: Review for clarity.
Lines 293-302: All species scientific names should be in Italic.
Lines 314-318: Authors mention that MSTRG.17494.1 lncRNA targets three plant NRT1/PTR family members. This statement is not supported by any specific experimental data shown in Results, except in silico prediction. Please include a figure describing results for this specific case of mRNA-lncRNA interaction or edit the Discussion accordingly to reflect this interaction is only predicted and not yet demonstrated in vitro.
In the Discussion, to complement the hypothesis presented in Figure 8, authors could point to specific protein coding or non-coding genes (lncRNAs) that could be overexpressed or silenced to improve nitrogen deficiency tolerance in sesame seedlings. This would be a nice follow up study deriving from this current publication.
English is overall fine but I recommend the editing team to proofread it and correct minor errors besides those I listed to authors.
Author Response
Dear Editor,
Thank you so much for your hard work on our manuscript. According to the reviewer’s comments, we carefully revised the manuscript “Transcriptome analysis of sesame (Sesamum indicum L.) reveal the LncRNAs and mRNAs regulatory network responding to low nitrogen stress” as follows:
Reviewer comments:
Reviewer #2:
I congratulate the authors for their work and manuscript. Manuscript ID: ijms-2977186 "Transcriptome analysis of sesame (Sesamum indicum L.) reveal the LncRNAs and mRNAs regulatory network responding to low nitrogen stress by Zhang and collaborators address the important issue of nitrogen assimilation and stress. Using sesame as a crop of choice, the work involves transcriptome analysis of protein coding and non-coding genes. Network mapping assisted the identification of important genes involved in nitrogen metabolism. Interestingly, interacting pairs of protein coding and non-coding RNAs were also identified, showing trans-regulation. The manuscript is well written and methods detailed. Results are presented clearly and support discussion and conclusion. It should be interesting to IJMS readers providing novel information on nitrogen metabolism and low nitrogen tolerance. I have only minor suggestions listed below and will recommend acceptance after minor edits to give you the opportunity to work on them.
In the title, use "reveals" instead of "reveal".
Response: Thanks for the reviewer’s advice, we made the revision in the title.
line 24: suggestion "...factors were identified in response to low-nitrogen stress...".
Response: Thanks for the reviewer’s advice, we made the revision in line 24.
line 50-51: review sentence for clarity.
Response: Thanks for the reviewer’s advice, we made the revision in line 50-51.
line 303-305: review sentence for clarity.
Response: Thanks for the reviewer’s advice, we made the revision in line 304-307.
line 316: suggestion "...NPF4.6 (LOC105168767). A previous study...".
Response: Thanks for the reviewer’s advice, we made the revision in line 322.
Table 1: Authors mention on legend "For each line, different low-96 ercase letters indicate significant differences (P≤0.05) among the treatments and genotypes", but no letters are shown next to the values on Table 1, please check.
Response: We are sorry for not showing those results. Please find the lowercase letters indicate significant differences (P≤0.05) in the revised Table 1.
Figure 1: Add number of replicate measurements on legend, and include error bars for fold change measurements.
Response: We are sorry for not showing those results. Please find the error bars for fold change measurements in the revised Figure 1. And we have added number of replicate measurements on legend in line 127.
Lines 237-239: Review for clarity.
Response: Thanks for the reviewer’s advice, we made the revision in line 235-238.
Lines 293-302: All species scientific names should be in Italic.
Response: Thanks for the reviewer’s advice, we made the revision in line 294-303.
Lines 314-318: Authors mention that MSTRG.17494.1 lncRNA targets three plant NRT1/PTR family members. This statement is not supported by any specific experimental data shown in Results, except in silico prediction. Please include a figure describing results for this specific case of mRNA-lncRNA interaction or edit the Discussion accordingly to reflect this interaction is only predicted and not yet demonstrated in vitro.
Response: Thanks for the reviewer’s advice. In this section, we predicted the potential cis-and trans-targeting transcripts of DELs by calculating Pearson correlation coefficient, (| PCC | > 0.6, P < 0.05). These 172 cis-transcripts were regulated by 92 regulatory DELs. Among them, MSTRG.17494.1 lncRNA targets three plant NRT1/PTR family members. These results were showed in supplementary Table S5, line 116-118, marked in red.
In the Discussion, to complement the hypothesis presented in Figure 8, authors could point to specific protein coding or non-coding genes (lncRNAs) that could be overexpressed or silenced to improve nitrogen deficiency tolerance in sesame seedlings. This would be a nice follow up study deriving from this current publication.
Response: We are grateful for your good suggestion. Subsequently, we will select key genes, such as NPF4.6 (LOC105168767), MYB54 (LOC105157332), PAL(LOC105161317) and so on to verify gene function and participate in the molecular mechanism of nitrogen response, which can provide genetic resources for low nitrogen tolerance breeding. We made the revision in the discussion part in line 382-385.
